# Association of METS-IR index with Type 2 Diabetes: A cross-sectional analysis of national health and nutrition examination survey data from 2009 to 2018

Yisen Hou[1], Rui Li[1], Zhen Xu[1], Wenhao Chen[1], Zhiwen Li[1], Weirong Jiang[1], Yong Meng[1]*, Jianli Han[2]*

1 Department of Oncology Surgery, Xi'an No.3 Hospital, The Afliated Hospital of Northwest Universit, Xi'an, Shanxi, People's Republic of China, 2 Department of General Surgery, Shanxi Bethune Hospital, Shanxi Academy of Medical Sciences, Third Hospital of Shanxi Medical University, Taiyuan, Shanxi, People's Republic of China

* my176@126.com (YM); hjl13803456545@126.com (JH)

## Abstract

### Background

With rising global diabetes prevalence, precise early identification and management of diabetes risk are critical research areas. The metabolic score for insulin resistance (METS-IR), a novel non-insulin-based tool, is gaining attention for quantifying insulin resistance using multiple metabolic parameters. Despite its potential in predicting diabetes and its precursors, evidence on its specific relationship with diabetes is limited, especially in large-scale population validation and mechanistic exploration.

### Objective

This study aims to analyze the association between METS-IR and type 2 diabetes (T2DM) in American adults.

### Methods

We conducted a cross-sectional analysis of the National Health and Nutrition Examination Survey (NHANES) data from 2009 to 2018. Participants aged 20 years and above were included, excluding individuals with missing data on BMI, fasting blood glucose, high-density lipoprotein cholesterol glycated hemoglobin and diabetes status. Logistic regression analysis, subgroup analysis, and restricted cubic spline analysis were used to assess the association between METS-IR and T2DM, controlling for potential confounding factors.

### Results

After adjusting for age, gender, race, education level, smoking status, drinking habits, depression, physical activity, hypertension, and hyperlipidemia, we found a positive association between METS-IR and the risk of T2DM. Specifically, each unit increase in METS-IR

**Data Availability Statement:** The dataset used to support the findings of this study is available

through the National Health and Nutrition Examination Survey (NHANES) repository at https://www.cdc.gov/nchs/nhanes/index.htm.

**Funding:** The author(s) received no specific funding for this work.

**Competing interests:** The authors have declared that no competing interests exist.

was associated with a 7% increase in the risk of T2DM (OR = 1.07, 95% CI: 1.06, 1.08). Subgroup analysis showed that the association between METS-IR and T2DM incidence was significantly positive in the highest quartile group, particularly among Mexican Americans over 40 years old and those diagnosed with depression, hypertension, or hyperlipidemia.

## Conclusion

Our study revealed a significant positive association between METS-IR and the prevalence of T2DM, indicating that this relationship persists even after controlling for various confounding factors. Therefore, monitoring METS-IR may provide a valuable tool for the early identification of individuals at risk of glucose metabolism disorders. Further research should focus on the applicability of METS-IR in different populations and its potential impact on clinical practice.

## 1. Introduction

In recent years, with the sharp increase in the global prevalence of diabetes, particularly Type 2 Diabetes Mellitus (T2DM), early identification and intervention have become significant challenges in public health [1]. T2DM is a complex metabolic disorder characterized by insulin resistance (IR), which reduces the body's sensitivity to insulin, resulting in dysregulated blood glucose control [2–4]. However, traditional methods to assess IR, such as the hyperinsulinemic-euglycemic clamp test [5,6], are complex and costly, limiting their applicability in clinical and large-scale population studies. The homeostasis model assessment of insulin resistance (HOMA-IR) index [7,8], widely used as an indirect method, is susceptible to variations in insulin measurement accuracy and lacks robust reproducibility. Therefore, there is a critical need to identify a simple, economical, and effective IR assessment tool.

Metabolic score for insulin resistance (METS-IR) has emerged as a novel, non-insulin-based metric for evaluating IR and has gained attention from researchers in recent years [9–13]. METS-IR integrates multiple metabolic parameters, including fasting glucose (FBG), triglycerides (TG), body mass index (BMI), and high-density lipoprotein cholesterol (HDL-C), into a formula, $Ln [(2 \times FBG + TG) \times BMI]/ Ln(HDL-C)$, to quantify IR severity [13]. It offers simplicity, cost-effectiveness, and potential applicability across diverse populations. While initial studies have shown associations between METS-IR and metabolic disorders like hypertension [14,15] and cardiovascular disease (CVD) [16,17], further exploration is needed to understand its predictive value in T2DM and its relationship with various demographic characteristics.

In the United States, a country with a high prevalence of diabetes [4], the National Health and Nutrition Examination Survey (NHANES) provides valuable resources for assessing the relationship between METS-IR and T2DM. NHANES, a large-scale survey tracking the health status of the U.S. population, includes extensive health indicators and demographic information, establishing a robust data foundation for investigating interrelationships among different health metrics.

This study aims to analyze the association between METS-IR and T2DM using NHANES data from 2009 to 2018. Through logistic regression analysis, subgroup analysis, and restricted cubic spline analysis, we aim to elucidate the potential of METS-IR in assessing T2DM risk

and explore its performance across diverse demographic characteristics such as age, gender, race, depression, hypertension, and hyperlipidemia. The findings from this study will not only deepen our understanding of the role of IR in the development of T2DM but also provide new insights and methodologies for early identification and intervention strategies for T2DM.

## 2. Materials and methods

### 2.1 Study design and population

The National Health and Nutrition Examination Survey is a continuous cross-sectional survey conducted by the National Center for Health Statistics (NCHS), a part of the Centers for Disease Control and Prevention (CDC) in the United States. NHANES aims to assess the health and nutritional status of both adults and children in the U.S. population by collecting data through questionnaires, physical examinations, and laboratory tests. It covers a wide range of health indicators, including chronic disease prevalence, nutritional status, dietary habits, health behaviors, and environmental exposures. For more detailed information, visit NHANES website (https://www.cdc.gov/nchs/nhanes/index.htm).

This study utilized NHANES data from 2009 to 2018 to explore the relationship between METS-IR index and the risk of T2DM, initially including a total of [initial number] participants. Participants were selected based on age criteria (>20 years old) and availability of METS-IR index data. Exclusion criteria were applied, including participants younger than 20 years old, those lacking METS-IR index data, and those without T2DM status information. After rigorous screening, a total of 14,699 eligible participants were included, among whom 3040 reported a history of T2DM (Fig 1).

### 2.2 METS-IR data collection

In this study, the METS-IR index was defined as the exposure variable. Levels of triglycerides, fasting glucose, and high-density lipoprotein cholesterol were measured using an automated biochemical analyzer [8]. Serum triglycerides and high-density lipoprotein cholesterol concentrations were determined using the Roche Cobas 6000 biochemical analyzer and Roche Modular P analyzer. BMI was calculated by directly measuring the participants' height and weight, using the formula BMI = weight (kg) / (height (m))$^2$. The formula for calculating the METS-IR index is the same as previously mentioned.

### 2.3 Type 2 Diabetes Mellitus (T2DM) data collection

Data on Type 2 Diabetes Mellitus were collected through a combination of laboratory measurements, questionnaires, and medical examinations. Fasting glucose and glycated hemoglobin (HbA1c) levels were measured using automated biochemical analyzers and high-performance liquid chromatography, respectively. Questionnaires provided information on participants' history of diabetes diagnosis and treatment. Medical examinations contributed basic physical indicators. The American Diabetes Association (ADA) 2022 criteria for diabetes diagnosis were referenced, alongside relevant information from the questionnaire regarding diabetes [18]. Participants were included in the Type 2 Diabetes Mellitus group if they met any of the following criteria: 1. Fasting glucose level $\geq$ 126 mg/dL (7.0 mmol/L); 2. HbA1c level $\geq$ 6.5% (48 mmol/mol); 3. DIQ010—Doctor told you have diabetes, participant answered "yes"; 4. DIQ050—Taking insulin now, participant answered "yes"; 5. DIQ070—Take diabetic pills to lower blood sugar, participant answered "yes".

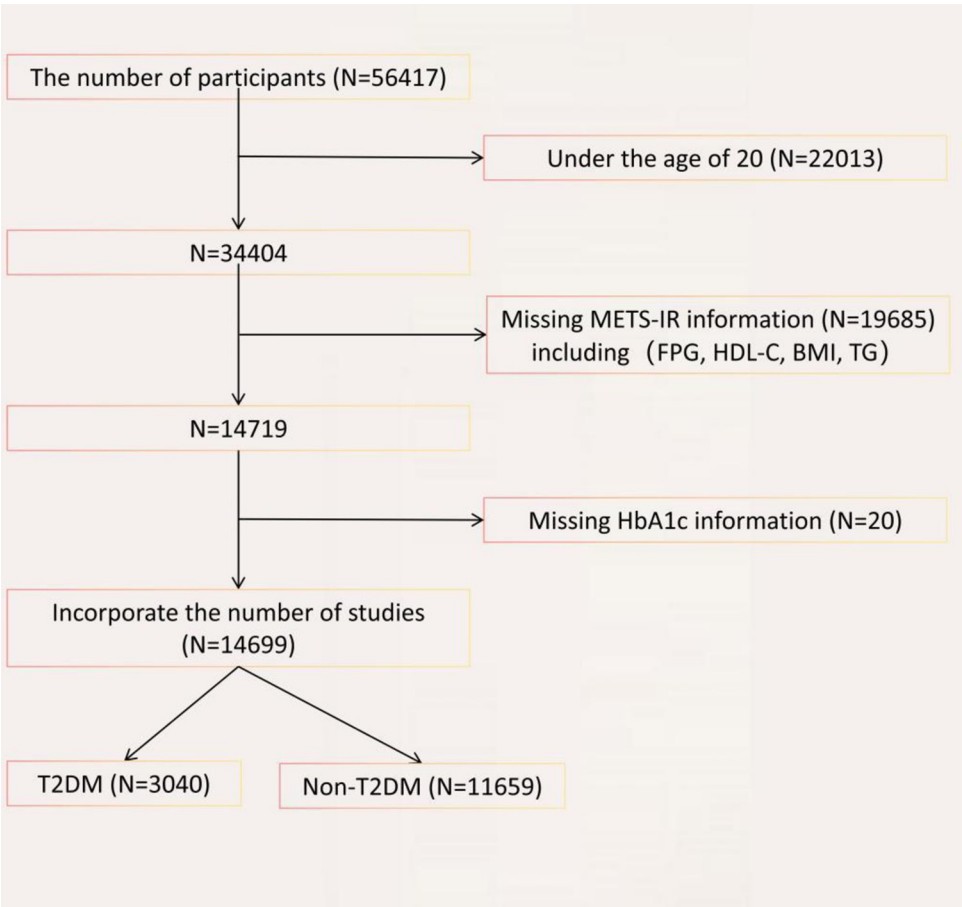

**Fig 1. Flowchart of participant selection.** A total of 56,417 participants were included, and after exclusions, the final number of diabetic patients was 3,040, and the number of non-diabetic individuals was 11,659.

## 2.4 Covariate selection

In this study, we selected a series of covariates to control for potential confounding factors. These covariates include age, gender, race, marital status, education level, poverty income ratio (PIR), smoking status, alcohol consumption habits, physical activity, hypertension status, hyperlipidemia status, and depressive symptoms. These variables were chosen based on a comprehensive consideration of existing literature and factors that could influence the relationship between METS-IR index and T2DM. Age was categorized into three groups: 20–40 years, 41–60 years, and >60 years. Poverty income ratio was defined as non-poor (PIR≥1) and poor (PIR<1). Hypertension was defined as systolic blood pressure ≥130 mmHg or diastolic blood pressure ≥80 mmHg [19]. The PHQ-9 questionnaire was used to assess the severity of depressive symptoms in the past two weeks, with each of the 9 questions scored from 0–3; a total score of 10 or higher indicated depression [20]. The Physical Activity Questionnaire (PAQ) was used to assess physical activity in the past week [21]. MET values for each activity were multiplied by its duration and summed to calculate the total MET for each participant. A total MET value greater than 600 was defined as normal physical activity.

## 2.5 Statistical analysis

To explore the association between METS-IR and the occurrence of Type 2 Diabetes Mellitus (T2DM), this study relies on the National Health and Nutrition Examination Survey

(NHANES), a large-scale, complex, multi-stage sampling survey. We strictly adhere to NHANES sampling weights and adjust for stratification and clustering effects to ensure robustness and validity of the statistical results.

Firstly, we conducted detailed descriptive statistical analysis of participants' baseline characteristics. Continuous variables were described using weighted means and their standard errors for precision, while categorical variables were presented with observed frequencies and weighted percentages to depict the distribution across different categories in the sample. Regarding the relationship between METS-IR and T2DM occurrence, we constructed three progressively adjusted models. Model 1 served as the baseline model without any adjustments. Model 2 adjusted for basic factors including age, gender, race, marital status, education level, and poverty income ratio (PIR). Model 3 comprehensively adjusted for all variables and specifically explored the trend effects of METS-IR quartiles on T2DM risk. We quantified the strength of association between METS-IR levels and T2DM risk using odds ratios (ORs) and their 95% confidence intervals (CIs). Additionally, restricted cubic spline analysis was employed to delve into the specific functional form of the relationship between METS-IR and T2DM incidence, capturing potential non-linear relationships between variables. Furthermore, subgroup analyses were conducted to investigate heterogeneity in the relationship between METS-IR and T2DM risk across key factors such as gender, age, race, depressive status, hypertension, and hyperlipidemia. Interaction tests were performed to assess whether significant interactions exist between METS-IR and these subgroup characteristics, aiming to deepen understanding of the pathways and mechanisms through which METS-IR influences T2DM risk. All statistical analyses were performed using R software version 4.4.0 (http://www.R-project.org, The R Foundation).

## 2.6 Ethical statement

Human subjects involved in the studies were granted ethical approval by the Ethical Review Board of the National Center for Health Statistics. The studies were conducted in compliance with the Declaration of Helsinki, local laws, and institutional guidelines. All participants provided Consent/Assent and Parental Permission for Specimen Storage and Continuing Studies, HOME INTERVIEW CONSENT, and CONSENT/ASSENT AND PARENTAL PERMISSION FOR EXAMINATION AT THE MOBILE EXAMINATION CENTER. All informed consent forms are publicly available on the NHANES official website.

# 3. Results

## 3.1 General characteristics of the study population

Baseline demographic characteristics of the study population are presented in Table 1, stratified by quartiles of METS-IR levels among 14,699 participants. The METS-IR index ranges for quartiles 1 to 4 were 17.14–34.00, 34.00–41.48, 41.48–50.45, and 50.45–132.24, respectively. Across different METS-IR quartile groups, there were notable differences in all factors except for smoking status and normal physical activity. Compared to quartile 1, participants in quartiles 2–4 were older. Quartile 4 exhibited higher proportions of males, Mexican Americans, individuals with below high school education, alcohol consumers, and those experiencing depressive symptoms. Total PA MET-minutes/week significantly decreased in quartile 4. Additionally, quartiles 2–4 showed a higher prevalence of diabetes, hypertension, and hyperlipidemia compared to quartile 1.

## 3.2 Relationship between METS-IR index and T2DM

Table 2 presents the association between METS-IR index and the risk of developing Type 2 Diabetes Mellitus (T2DM). Participants were categorized into quartiles based on their

**Table 1. Weighted baseline characteristics of the participants.**

| Characteristic | Q1 [17.14, 34.00] | Q2 [34.00, 41.48] | Q3 [41.48, 50.45] | Q4 [50.45, 132.24] | P-value |
|---|---|---|---|---|---|
| N | 3458 | 3712 | 3776 | 3753 | |
| METS-IR | 29.42(3.09) | 37.74(2.12) | 45.64(2.57) | 61.29(10.27) | <0.001 |
| Age (years) | 44.82(17.93) | 49.44 (17.10) | 49.80(16.53) | 48.27(15.71) | <0.001 |
| 20–40 (%) | 46.2% | 34.7% | 31.6% | 35.2% | |
| 41–60 (%) | 31.9% | 35.9% | 41.2% | 39.7% | |
| >60 (%) | 21.9% | 29.5% | 27.2% | 25.2% | |
| Gender (%) | | | | | <0.001 |
| Male | 36.5% | 50.6% | 56.2% | 50.7% | |
| Female | 63.5% | 49.4% | 43.8% | 49.3% | |
| Race (%) | | | | | <0.001 |
| Mexican American | 5.0% | 8.0% | 10.8% | 11.4% | |
| Non-Hispanic White | 67.8% | 66.2% | 63.8% | 64.4% | |
| Non-Hispanic Black | 9.5% | 10.1% | 10.5% | 11.9% | |
| Other Race | 17.7% | 15.7% | 14.9% | 12.2% | |
| Married/live with partner (%) | | | | | 0.045 |
| Yes | 60.0% | 64.4% | 65.1% | 65.1% | |
| No | 40.0% | 35.6% | 34.9% | 34.9% | |
| Education level (%) | | | | | <0.001 |
| Below high school | 11.6% | 15.0% | 16.9% | 17.0% | |
| High School or above | 88.4% | 85.0% | 83.1% | 83.0% | |
| Poverty income ratio (%) | | | | | 0.044 |
| Poor | 12.6% | 11.8% | 13.8% | 15.7% | |
| Not poor | 78.8% | 80.4% | 78.4% | 76.6% | |
| Unclear | 8.7% | 7.8% | 7.9% | 7.7% | |
| Smoking (%) | | | | | 0.214 |
| Yes | 17.3% | 15.3% | 14.6% | 14.6% | |
| No | 82.7% | 84.7% | 85.4% | 85.4% | |
| Alcohol (%) | | | | | 0.003 |
| Yes | 84.7% | 82.5% | 84.1% | 80.7% | |
| No | 15.3% | 17.5% | 15.9% | 19.3% | |
| PHQ-9 | 2.84(3.90) | 2.74(3.76) | 3.18(4.29) | 3.74(4.55) | <0.001 |
| Depress(%) | | | | | <0.001 |
| Yes | 7.1% | 6.1% | 8.5% | 11.1% | |
| No | 92.9% | 93.9% | 91.5% | 88.9% | |
| Total PA MET-minutes/week | 623.15 (1856.50) | 648.31 (2192.46) | 591.14 (1654.13) | 515.51 (1114.81) | 0.005 |
| Met PA (%) | | | | | 0.434 |
| Yes | 27.7% | 26.2% | 26.2% | 25.1% | |
| No | 72.3% | 73.8% | 73.8% | 74.9% | |
| Diabetes (%) | | | | | <0.001 |
| Yes | 3.5% | 8.8% | 16.8% | 12.6% | |
| No | 96.5% | 91.2% | 83.2% | 67.4% | |
| Hypertension (%) | | | | | <0.001 |
| Yes | 29.3% | 43.2% | 55.1% | 66.6% | |
| No | 70.7% | 56.8% | 44.9% | 33.4% | |
| Hyperlipidaemia (%) | | | | | <0.001 |
| Yes | 50.4% | 71.6% | 78.7% | 86.9% | |
| No | 49.6% | 28.4% | 21.3% | 13.1% | |

Table 2. Weighted association between METS-IR index and T2DM.

| Characteristic | OR (95% CI), P-value | | |
|---|---|---|---|
| | Model 1 | Model 2 | Model 3 |
| METS-IR | 1.07(1.06, 1.07) <0.001 | 1.08(1.07, 1.08) <0.001 | 1.07(1.06, 1.08) <0.001 |
| Quartile 1 | 1.00 (reference) | 1.00 (reference) | 1.00 (reference) |
| Quartile 2 | 2.66 (2.05,3.44) <0.001 | 2.15 (1.65,2.79), <0.001 | 2.00 (1.47,2.72), <0.001 |
| Quartile 3 | 5.54 (10.47,16.84) <0.001 | 4.74 (3.68,6.10), <0.001 | 4.14 (3.09,5.56), <0.001 |
| Quartile 4 | 13.28(2.47,5.34) <0.001 | 14.17 (10.99,18.28), <0.001 | 11.25 (8.40,15.08), <0.001 |
| p for trend | 1.09(1.08, 1.09) <0.001 | 1.10(1.09, 1.11) <0.001 | 1.09(1.08, 1.10) <0.001 |

OR, odds ratio; CI, confidence interval.

**Model 1:** Crude model.

**Model 2:** Adjusted for age, gender, race, marital status, education level, and poverty income ratio.

**Model 3:** Adjusted for age, gender, race, educational level, poverty income ratio, sleep hours, smoking, alcohol consumption, Depress, Metabolic Physical Activity and history of hypertension, and hyperlipidemia.

METS-IR index, with the lowest quartile (quartile 1) designated as the reference group. In both crude and fully adjusted models, there was a positive correlation between METS-IR index and the prevalence of T2DM. The fully adjusted model indicated that for every unit increase in METS-IR index, the risk of T2DM increased by 7% (OR = 1.07, 95% CI: 1.06–1.08).

Furthermore, sensitivity analysis was conducted by categorizing METS-IR index into quartiles (Q1-Q4) to explore its effect as a categorical variable. In Model 3, adjusting for multiple covariates including poverty income ratio (PIR), smoking status, alcohol consumption habits, physical activity, hypertension status, hyperlipidemia status, and depressive symptoms, participants in quartiles 2 to 4 had significantly higher risks of T2DM compared to quartile 1. Specifically, the risks were 2.00 times higher (OR = 2.00, 95% CI: 1.47–2.72) for quartile 2, 4.14 times higher (OR = 4.14, 95% CI: 3.09–5.56) for quartile 3, and 11.25 times higher (OR = 11.25, 95% CI: 8.40–15.08) for quartile 4.

Multivariable-adjusted spline curves depicting the relationship between METS-IR index and T2DM prevalence are shown in Fig 2. Higher METS-IR index values were linearly associated with increased prevalence of T2DM (non-linearity p-values >0.05).

### 3.3 Subgroup analysis

Subgroup analysis was conducted to assess whether the relationship between METS-IR index and the prevalence of Type 2 Diabetes Mellitus (T2DM) is influenced by age, gender, race, depressive status, hypertension, and hyperlipidemia. As shown in Fig 3, stratification factors included gender, age, race, depressive status, hypertension, and hyperlipidemia. We found that race modified the association between METS-IR and T2DM (interaction p < 0.05). However, age, gender, depressive status, hypertension, and hyperlipidemia did not significantly alter the correlation between METS-IR and T2DM (interaction p > 0.05).

### 4. Discussion

In this study, we utilized national health and nutrition survey data to investigate the association between METS-IR Index and T2DM. We also explored whether these associations are moderated by factors such as age, gender, race, depressive status, hypertension, and

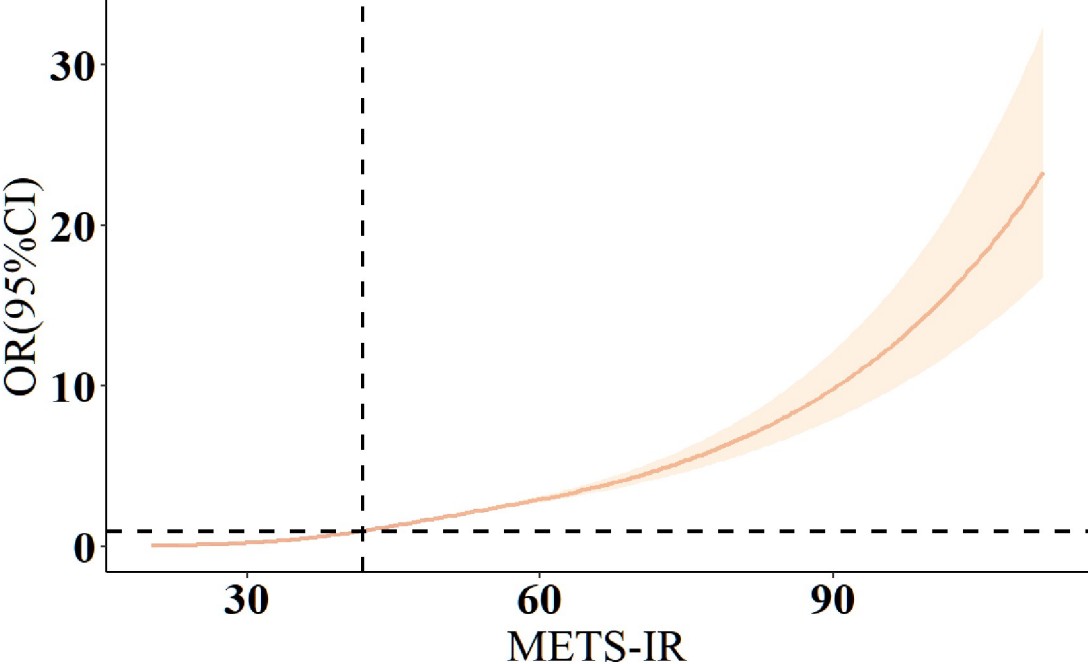

**Fig 2. Restricted cubic spline analysis of the association between METS-IR index and Type 2 Diabetes Mellitus risk.** We conducted restricted cubic spline analysis to assess the association between METS-IR index and the risk of developing Type 2 Diabetes Mellitus (T2DM). The model was adjusted for all covariates. Odds ratios (OR) were depicted using a solid orange line, while the 95% confidence intervals were represented by a lightly shaded yellow area.

hyperlipidemia. The study found that individuals with higher METS-IR indices have a significantly increased risk of T2DM. For each unit increase in METS-IR, there was a 7% higher risk of T2DM. Even after categorizing METS-IR from a continuous variable to a categorical variable, it remained significantly associated with T2DM prevalence. Furthermore, subgroup analysis indicated a significant interaction between METS-IR and race, suggesting varying risks of developing diabetes under the influence of metabolic syndrome across different racial groups.

Our study results align with previous research. For instance, Cheng Hui et al. [22] found a significant association between high METS-IR index and the incidence of Type 2 Diabetes Mellitus (T2DM) (OR: 1.804; 95% CI: 1.720–1.891). However, their study sample consisted of the Chinese population and did not account for glycated hemoglobin levels or prior use of insulin and oral hypoglycemic agents in the included population. Our study further revealed the moderating effect of race on this association, which has not been sufficiently explored in previous research. Possible reasons include differences in dietary habits and gut microbiota among different racial groups, leading to varying impacts on metabolism and insulin sensitivity. This requires further validation in future studies. In another study [13], they also confirmed a significant association between increased baseline METS-IR and the incidence of T2DM. They found that participants in the highest baseline METS-IR quartile had a 2.72 times higher adjusted risk of developing new-onset T2DM compared to those in the lowest METS-IR quartile. This is consistent with our study, but we found that the risk of incidence in quartile 4 was more than ten times higher compared to quartile 1. The reason for this discrepancy may be related to higher METS-IR values in our quartile groups and different confounding factors included. Similarly, studies by Bello-Chavolla et al. [23] and Ming Zhang et al. [10] also demonstrated a significant association between the METS-IR index and the incidence of T2DM, despite differences in risk magnitudes.

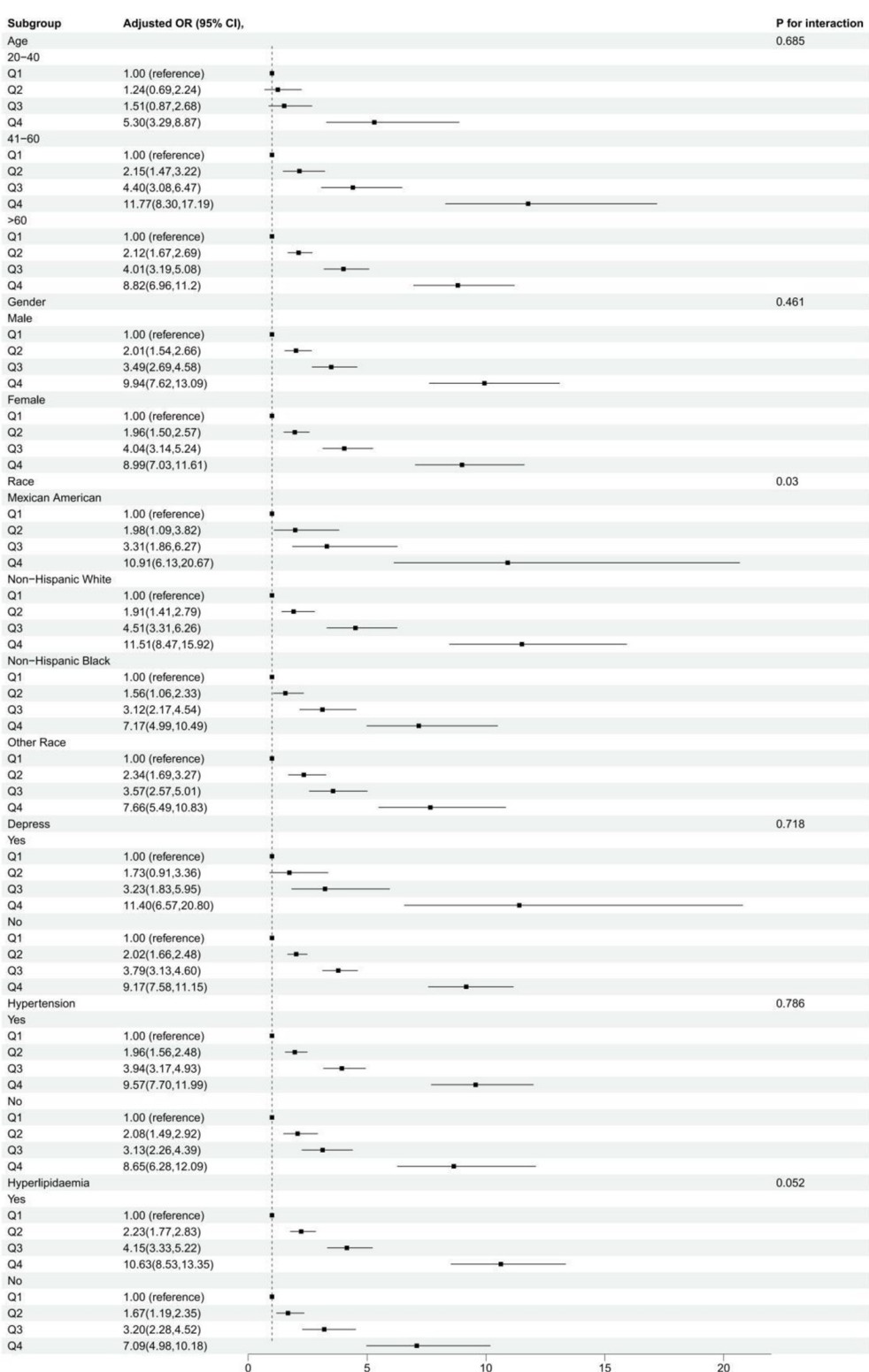

**Fig 3. Subgroup analysis of METS-IR index and Type 2 Diabetes Mellitus prevalence.** This figure presents the results of subgroup analysis evaluating the influence of demographic and health factors on the relationship between METS-IR index and the prevalence of T2DM. Stratification factors included gender, age, race, depressive status, hypertension, and hyperlipidemia.

The association between the METS-IR Index and T2DM involves complex mechanisms across various pathophysiological processes. Firstly, insulin resistance is a core characteristic of T2DM [2]. A high METS-IR index indicates significant insulin resistance, leading to difficulties in lowering blood glucose levels. Studies have shown that METS-IR is negatively correlated with insulin sensitivity, and higher METS-IR levels are associated with impaired β-cell function, further exacerbating hyperglycemia [23]. Secondly, chronic inflammation plays a crucial role in the development of T2DM [24]. Inflammatory factors such as IL-6 and TNF-α inhibit insulin signaling pathways, reducing insulin sensitivity [25]. Research indicates that the METS-IR index is positively correlated with levels of inflammatory markers like CRP and IL-6, suggesting that higher insulin resistance corresponds to stronger inflammatory responses [26]. Obesity and visceral fat accumulation, which promote the release of inflammatory factors, are significant contributors to chronic inflammation. Disrupted lipid metabolism is another critical factor in the development of T2DM. Dysfunction in adipocytes leads to increased free fatty acids and inflammation within adipose tissue, interfering with insulin signaling [27,28]. Additionally, mitochondrial dysfunction affects cellular energy metabolism, increases oxidative stress, and disrupts insulin signaling [29]. The interaction between genes and the environment also plays an essential role in the pathogenesis of T2DM [30]. Individuals exhibit varying sensitivities to environmental factors such as diet and exercise, with multiple genetic variations closely linked to T2DM risk. For instance, TCF7L2 and PPARG gene variations affect insulin secretion and sensitivity [30,31]. The FTO gene variation is associated with obesity and influences appetite regulation and energy balance, increasing the risk of T2DM [32]. Environmental factors, including diet and physical activity, can alter gene expression through epigenetic mechanisms, further impacting insulin resistance and T2DM risk. In conclusion, the mechanisms linking METS-IR and T2DM include insulin resistance, inflammatory responses, disrupted lipid metabolism, mitochondrial dysfunction, and gene-environment interactions. These mechanisms collectively influence the onset and progression of T2DM.

This study has several strengths. Firstly, we used nationally representative large-scale data, ensuring high representativeness and generalizability of the results. Secondly, we employed multivariable adjustment models to control for potential confounding factors as much as possible. However, there are some limitations. Firstly, the cross-sectional study design does not allow for the determination of causal relationships. Secondly, the data were self-reported, which may introduce reporting bias. Additionally, the calculation of the METS-IR index depends on various metabolic indicators, which can be influenced by multiple factors, leading to increased variability in the METS-IR index. Lastly, the study sample primarily comprised individuals from the United States, so the results may not be generalizable to other populations. Future research could adopt a longitudinal study design to establish the causal relationship between METS-IR and type 2 diabetes. Furthermore, similar studies in different populations are needed to validate our findings and assess their generalizability. Moreover, further research into the interactions between METS-IR and other metabolic indicators, as well as how these interactions manifest in different populations, could enhance our understanding of the impact of metabolic syndrome on diabetes risk.

## 5. Conclusion

This study utilized the NHANES database to investigate the association between METS-IR and T2DM. Our results indicate that after adjusting for potential confounding factors, METS-IR is significantly associated with T2DM. Subgroup analysis revealed that race significantly influences the association between METS-IR and T2DM, while the effects of age, gender,

depression status, hypertension, and hyperlipidemia were not significant. This study provides important evidence supporting METS-IR as an effective indicator for assessing T2DM risk. Measuring and monitoring METS-IR can help identify high-risk individuals early, allowing for effective intervention and management before the disease manifests. This is valuable for reducing the burden of T2DM, decreasing the incidence of related complications, and improving patients' quality of life. Future research should further explore the pathophysiological mechanisms of METS-IR and its potential intervention strategies to better prevent and manage T2DM. Additionally, research on different populations is also crucial. In summary, this study not only reveals the significant association between METS-IR and T2DM risk but also points to directions for future research and clinical practice. By further exploring the mechanisms of METS-IR and its intervention strategies, we hope to achieve greater progress in preventing and managing T2DM.

## Author Contributions

**Conceptualization:** Yisen Hou, Rui Li, Yong Meng, Jianli Han.

**Data curation:** Yisen Hou, Rui Li, Wenhao Chen, Jianli Han.

**Formal analysis:** Yong Meng.

**Methodology:** Yisen Hou, Rui Li, Wenhao Chen, Yong Meng, Jianli Han.

**Resources:** Yisen Hou.

**Software:** Yisen Hou, Zhen Xu, Wenhao Chen, Zhiwen Li, Weirong Jiang, Jianli Han.

**Supervision:** Zhen Xu, Zhiwen Li, Weirong Jiang, Yong Meng, Jianli Han.

**Validation:** Zhiwen Li, Weirong Jiang, Yong Meng, Jianli Han.

**Visualization:** Yisen Hou, Zhen Xu, Wenhao Chen, Zhiwen Li, Weirong Jiang, Yong Meng, Jianli Han.

**Writing – original draft:** Yisen Hou.

**Writing – review & editing:** Yisen Hou, Rui Li, Zhen Xu, Wenhao Chen, Zhiwen Li, Weirong Jiang, Yong Meng, Jianli Han.

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
