## [Decision Letter · Decision Letter 0]

15 Sep 2024

PONE-D-24-31087Association of METS-IR Index with Type 2 Diabetes: A Cross-sectional Analysis of National Health and Nutrition Examination Survey Data from 2009 to 2018PLOS ONE

Dear Dr. Han,

Thank you for submitting your manuscript to PLOS ONE. After careful consideration, we feel that it has merit but does not fully meet PLOS ONE’s publication criteria as it currently stands. Therefore, we invite you to submit a revised version of the manuscript that addresses the points raised during the review process.

We look forward to receiving your revised manuscript.

Kind regards,

Fredirick Lazaro mashili, MD, PhD

Academic Editor

PLOS ONE

Journal Requirements:

Additional Editor Comments: 

Please address thoroughly the minor comments raised by the reviewer. Remmember to review and follow all the journal’s formatting and stying requirements when submitting the revised version.Please address sufficiently all the concerns raised by the reviewersThis manuscript requires a minor revisionCarefull review all the journa’s formatting and styling requirements and make sure to adhere when submitting a revised manuscript.

Reviewers' comments:

Reviewer's Responses to Questions

**Comments to the Author**

1. Is the manuscript technically sound, and do the data support the conclusions?

Reviewer #1: Yes

Reviewer #2: Yes

2. Has the statistical analysis been performed appropriately and rigorously? 

Reviewer #1: Yes

Reviewer #2: Yes

3. Have the authors made all data underlying the findings in their manuscript fully available?

Reviewer #1: Yes

Reviewer #2: Yes

4. Is the manuscript presented in an intelligible fashion and written in standard English?

Reviewer #1: Yes

Reviewer #2: Yes

5. Review Comments to the Author

Reviewer #1: The study is highly significant and offers valuable insights, particularly in addressing non-communicable diseases such as Type 2 Diabetes. The manuscript is well-written and clearly articulated. The METS-IR metric merits further exploration to better understand its causal relationship with Diabetes Mellitus and other NCDs, given its simplicity and cost-effectiveness.

Comments:

1. In the Ethics Statement section, the author should specify whether the study was conducted in accordance with the `Helsinki Declaration`.

2. The term "METS-IR" should be written out in full the first time it appears, as should "NHANES."

Reviewer #2: In this study the authors aimed to determine the association between insulin resistance score (MET-IR) and T2DM. The author conclude that a significant association exist between MET-IR and T2DM both in crude and adjusted models. They used systematically acquired survey data, employed appropriate and robust methods and statistics to address their research question, and have presented their results with clarity and scientific soundness.

6. PLOS authors have the option to publish the peer review history of their article (what does this mean?). If published, this will include your full peer review and any attached files.

Reviewer #1: **Yes: **Ikunda Dionis

Reviewer #2: **Yes: **Fredirick mashili

---

## [Author Response · Author response to Decision Letter 0]

17 Sep 2024

We would like to express our sincere thanks to the reviewers for the constructive and positive comments.

Specific Comments

Comment 1: Please ensure that your manuscript meets PLOS ONE's style requirements, including those for file naming. 

Answer: Thank you for your reminder. We will carefully review and ensure that our manuscript adheres to PLOS ONE's style requirements, including the proper file naming conventions.

We appreciate your guidance and will make the necessary adjustments. 

Comment 2: PLOS requires an ORCID iD for the corresponding author in Editorial Manager on papers submitted after December 6th, 2016. Please ensure that you have an ORCID iD and that it is validated in Editorial Manager. To do this, go to ‘Update my Information’ (in the upper left-hand corner of the main menu), and click on the Fetch/Validate link next to the ORCID field. This will take you to the ORCID site and allow you to create a new iD or authenticate a pre-existing iD in Editorial Manager.

Answer: Thank you for the reminder. I confirm that I have an ORCID iD, and I will ensure that it is properly validated in the Editorial Manager by following the provided instructions.

I appreciate your guidance on this matter.

Comment 3: Your ethics statement should only appear in the Methods section of your manuscript. If your ethics statement is written in any section besides the Methods, please move it to the Methods section and delete it from any other section. Please ensure that your ethics statement is included in your manuscript, as the ethics statement entered into the online submission form will not be published alongside your manuscript. 

Answer: Thank you for your comment. We will move the ethics statement to the Methods section of the manuscript and ensure it is only included there. We will also make sure that the ethics statement is properly incorporated into the manuscript, as per the guidelines.

We appreciate your guidance on this matter.

Comment 4: Please review your reference list to ensure that it is complete and correct. If you have cited papers that have been retracted, please include the rationale for doing so in the manuscript text, or remove these references and replace them with relevant current references. Any changes to the reference list should be mentioned in the rebuttal letter that accompanies your revised manuscript. If you need to cite a retracted article, indicate the article’s retracted status in the References list and also include a citation and full reference for the retraction notice. 

Answer: Thank you for your helpful comment. We will thoroughly review our reference list to ensure it is complete and correct. If any retracted papers have been cited, we will either remove them and replace them with relevant current references, or provide a rationale for their inclusion in the manuscript. We will detail any changes made to the reference list in the rebuttal letter that accompanies the revised manuscript.

Original references: [18] 2. Classification and Diagnosis of Diabetes: Standards of Medical Care in Diabetes-2021 [J]. Diabetes Care, 2021, 44(Suppl 1): S15-s33.

modified references: [18] American Diabetes Association. 2. Classification and Diagnosis of Diabetes: Standards of Medical Care in Diabetes-2021. Diabetes Care. 2021 Jan;44(Suppl 1):S15-S33. 

The original references were imported via EndNote, and the information regarding the authoring institutions was missing. After modification, this information has been added.

Comment 5: In the Ethics Statement section, the author should specify whether the study was conducted in accordance with the `Helsinki Declaration`. 

Answer: Thank you for your valuable suggestions on our manuscript. We confirm that this study was conducted in strict accordance with the Declaration of Helsinki. In the upcoming revisions, we will explicitly mention this in the Ethics Statement section to ensure compliance and transparency.

Comment 6: The term "METS-IR" should be written out in full the first time it appears, as should "NHANES."

Answer: Thank you for your insightful comment. We will ensure that the term "METS-IR" is written out in full as "Metabolic Score for Insulin Resistance" upon its first mention, as well as "NHANES" as "National Health and Nutrition Examination Survey."

---

## [Decision Letter · Decision Letter 1]

22 Oct 2024

Association of METS-IR Index with Type 2 Diabetes: A Cross-sectional Analysis of National Health and Nutrition Examination Survey Data from 2009 to 2018

PONE-D-24-31087R1

Dear Dr. Han,

We’re pleased to inform you that your manuscript has been judged scientifically suitable for publication and will be formally accepted for publication once it meets all outstanding technical requirements.

Kind regards,

Fredirick Lazaro mashili, MD, PhD

Academic Editor

PLOS ONE

Additional Editor Comments (optional):

All the comments have been sufficiently addressed.

Reviewers' comments:

Reviewer's Responses to Questions

**Comments to the Author**

1. If the authors have adequately addressed your comments raised in a previous round of review and you feel that this manuscript is now acceptable for publication, you may indicate that here to bypass the “Comments to the Author” section, enter your conflict of interest statement in the “Confidential to Editor” section, and submit your "Accept" recommendation.

Reviewer #1: All comments have been addressed

Reviewer #2: All comments have been addressed

2. Is the manuscript technically sound, and do the data support the conclusions?

Reviewer #1: Yes

Reviewer #2: Yes

3. Has the statistical analysis been performed appropriately and rigorously? 

Reviewer #1: Yes

Reviewer #2: Yes

4. Have the authors made all data underlying the findings in their manuscript fully available?

Reviewer #1: Yes

Reviewer #2: Yes

5. Is the manuscript presented in an intelligible fashion and written in standard English?

Reviewer #1: Yes

Reviewer #2: Yes

6. Review Comments to the Author

Reviewer #1: The manuscript is well-written, and the authors have thoroughly addressed all the comments. The study is original, with METS-IR demonstrating potential as a promising, cost-effective, and user-friendly tool for predicting diabetes mellitus. However, the findings related to racial variability require further investigation across different racial groups to draw more conclusive results.

Reviewer #2: All comments have been sufficiently addressed. The authors have provided all what was asked by the reviewers.

7. PLOS authors have the option to publish the peer review history of their article (what does this mean?). If published, this will include your full peer review and any attached files.

Reviewer #1: **Yes: **Ikunda Dionis

Reviewer #2: **Yes: **Fredirick mashili

---

## [Editor Report · Acceptance letter]

29 Oct 2024

PONE-D-24-31087R1 

PLOS ONE

Dear Dr. Han, 

I'm pleased to inform you that your manuscript has been deemed suitable for publication in PLOS ONE. Congratulations! Your manuscript is now being handed over to our production team.

Kind regards, 

on behalf of

Dr Fredirick Lazaro mashili 

Academic Editor

PLOS ONE